# A single, improbable B cell receptor mutation confers potent neutralization against cytomegalovirus

**Jennifer A. Jenks**[1], **Sharmi Amin**[1], **Madeline R. Sponholtz**[2], **Amit Kumar**[1], **Daniel Wrapp**[2], **Sravani Venkatayogi**[1], **Joshua J. Tu**[1], **Krithika Karthigeyan**[3], **Sarah M. Valencia**[1], **Megan Connors**[3], **Melissa J. Harnois**[1], **Bhavna Hora**[1], **Eric Rochat**[1], **Jason S. McLellan**[2], **Kevin Wiehe**[1,4⊙], **Sallie R. Permar**[1,3⊙]*

1 Duke Human Vaccine Institute, Duke University Medical Center, Durham, North Carolina, United States of America, 2 Department of Molecular Biosciences, The University of Texas at Austin, Austin, Texas, United States of America, 3 Department of Pediatrics, Weill Cornell Medicine, New York, New York, United States of America, 4 Department of Medicine, Duke University School of Medicine, Durham, North Carolina, United States of America

⊙ These authors contributed equally to this work.
* sallie.permar@med.cornell.edu

**Data Availability Statement:** The authors confirm that all data underlying the findings are fully available without restriction. All relevant data are

## Abstract

Cytomegalovirus (CMV) is a leading cause of infant hearing loss and neurodevelopmental delay, but there are no clinically licensed vaccines to prevent infection, in part due to challenges eliciting neutralizing antibodies. One of the most well-studied targets for CMV vaccines is the viral fusogen glycoprotein B (gB), which is required for viral entry into host cells. Within gB, antigenic domain 2 site 1 (AD-2S1) is a target of potently neutralizing antibodies, but gB-based candidate vaccines have yet to elicit robust responses against this region. We mapped the genealogy of B cells encoding potently neutralizing anti-gB AD-2S1 antibodies from their inferred unmutated common ancestor (UCA) and characterized the binding and function of early lineage ancestors. Surprisingly, we found that a single amino acid heavy chain mutation A33N, which was an improbable mutation rarely generated by somatic hypermutation machinery, conferred broad CMV neutralization to the non-neutralizing UCA antibody. Structural studies revealed that this mutation mediated key contacts with the gB AD-2S1 epitope. Collectively, these results provide insight into potently neutralizing gB-directed antibody evolution in a single donor and lay a foundation for using this B cell-lineage directed approach for the design of next-generation CMV vaccines.

## Author summary

Despite over 50 years of research, CMV vaccine candidates have achieved only up to 50% efficacy in clinical trials, in part due to challenges generating neutralizing antibody responses. One of the most promising targets is CMV gB, which mediates viral entry into host cells, and specifically the gB AD-2S1 epitope, which is a target of neutralizing antibodies in natural infection that have not yet been successfully elicited by vaccination.

within the paper and its Supporting information files.

**Funding:** This work was supported by NIH F30 to J.A.J. (F30-HD100170-01A1), Triangle Center for Evolutionary Medicine Graduate Student Award to J.A.J., National CMV Foundation Early Career Award to J.A.J., Welch Foundation grant to J.S.M. (F-0003-19620604), NIH R21 to S.R.P. (R21-AI147992-01), and Medearis CMV Scholars Program Award to J.A.J. The funders had no role in study design, data collection and interpretation, decision to publish, or preparation of this manuscript.

**Competing interests:** I have read the journal's policy and the authors of this manuscript have the following competing interests: J.A.J. has been a paid invited speaker by Moderna x Popsugar. S.R. P. serves as a consultant for Moderna, Merck, Dynavax, Pfizer, and Hookipa CMV vaccine programs and has a sponsored research program on CMV vaccine immunogenicity with Moderna and Merck. J.A.J., K.W., and S.R.P. submitted a provisional patent (#9878-01-US) for antibodies described in this manuscript.

Utilizing B cell lineage analysis of a neutralizing gB AD-2S1-specific monoclonal antibody lineage, we identified a single, improbable heavy chain mutation that conferred neutralizing function and mediated a key contact within the epitope. Our study suggests that lineage-based vaccine design may be used to target induction of CMV gB AD-2S1-specific potently neutralizing antibodies.

## Introduction

Human cytomegalovirus (CMV, human herpesvirus 5) is a pervasive viral pathogen and major cause of disease in infants and immunocompromised patients worldwide [1, 2]. Although CMV infection is typically asymptomatic in healthy adults, CMV transmission *in utero* can cause permanent hearing loss, cognitive impairment, retinitis, and cerebral palsy in affected infants [3]. Congenital CMV infection alone is responsible for nearly a quarter of all newborn hearing loss [4]. CMV infection is also a major cause of morbidity and mortality in transplant recipients and persons with HIV [5]. Accordingly, there have been many efforts to develop vaccines that will prevent infection and transmission with the goal of reducing the global CMV-related burden of disease [6]. However, vaccine development has faced challenges identifying immunogens that can induce broad and potent immunity and confer sustained protection against CMV infection [7].

One of the leading targets for vaccine development is CMV glycoprotein B (gB), which is a viral envelope protein that mediates fusion with host-cell membranes and is required for viral entry into all known cell types [8, 9]. Indeed, the most efficacious CMV vaccine trialed to date was composed of a postfusion CMV gB subunit protein combined with an MF59 adjuvant (gB/MF59) that conferred approximately 50% protection from primary acquisition in multiple phase 2 clinical trials [10–13]. Follow-up immunogenicity studies found that in CMV-seronegative vaccinees, gB/MF59 elicited antibody responses against three of the four total neutralizing antigenic domains of gB [14, 15] but not against gB antigenic domain-2 site 1 (AD-2S1), which is known to be the target of potently neutralizing antibodies in natural infection [16].

CMV gB AD-2S1 is a linear epitope at the N-terminus of gB (amino acids 68–81), which is extracellular, and is highly conserved across clinical strains [17]. Multiple studies have implicated antibodies against gB AD-2S1 in protection from CMV disease and vertical transmission. In CMV-seropositive renal transplant recipients, the presence of serum anti-gB AD-2S1 antibodies was associated with reduced risk of post-transplant CMV disease [18]. In CMV-seropositive transplant recipients immunized with gB/MF59, vaccination boosted anti-gB AD-2S1 serum antibody titers in a subset of subjects with preexisting anti-gB AD-2S1 antibodies, and these titers were associated with protection from CMV viremia [19]. We also found that in a study of *in utero* CMV transmission among HIV-infected mothers, the presence of maternal serum antibodies against gB AD-2S1 was associated with a reduced risk of vertical CMV transmission [20]. As elicited in the context of natural infection, antibodies targeting gB AD-2S1 are well recognized for their role in protection from CMV viremia and congenital transmission. Although the function of the AD-2S1 region in gB not yet fully understood, a recently published crystallographic structure of the prefusion gB suggested that the CMV gB AD-2S1 region was responsible for binding to gH/gL in the pentameric complex and that this interaction may have a role in triggering gB-mediated cell entry [21, 22].

Based on its location at the extracellular N-terminus, gB AD-2S1 should be readily available for immune recognition. Yet, surprisingly, only ~50% of naturally CMV-infected individuals have detectable circulating anti-gB AD-2S1 antibodies [23]. Vaccine candidates to

date including gB/MF59 have failed to elicit antibodies targeting this region [14, 15], and direct immunization with gB AD-2S1 peptides in animal models has also failed to elicit neutralizing antibody responses [23]. These findings indicate the presence of barriers, such as structural constraints within gB and host genetic restriction, that prevent the generation of neutralizing antibodies targeting gB AD-2S1. Indeed, there are two known glycosylation sites within gB AD-2S1 that may contribute to glycan shielding, and crystal structures of postfusion gB suggest that the nearby gB antigenic domain-1 may cloak the gB AD-2S1 region, blocking immune access to this epitope [17].

The host genetic restriction of germline B cells may also prevent the generation of neutralizing anti-gB AD-2S1 antibodies. Germline B cell receptors can have thousands of potential heavy ($V_H$) and light ($V_L$) pairings, yet all of the potently neutralizing gB AD-2S1 monoclonal antibodies (mAbs) isolated from naturally infected individuals to date are derived from the same $V_H/V_L$ pairing: IGHV3-30 (or the related IGHV3-30-3) and IGKV3-11 [24–26]. The use of a single pairing out of thousands of potential combinations reflects the incredibly small subset of germline B cells with complementarity-determining regions (CDRs) capable of recognizing gB AD-2S1. Moreover, non-germline-encoded residues may be critical for high binding to gB AD-2S1 [27], presenting a mutational barrier to the development of these neutralizing lineages. Thus, the elicitation of gB AD-2S1-specific antibodies by vaccination likely requires rational vaccine design that addresses potential germline restriction and overcomes barriers in lineage maturation.

B cell lineage-targeted vaccine design aims to direct germline B cells along favorable maturation pathways, such as those targeting gB AD-2S1. In this approach, the germline B cell precursor sequence of clonally related antibodies with desired neutralizing potency is inferred using phylogenetic methods, then the amino acid mutations associated with the development of neutralizing functions along lineage maturation are identified. Antibodies with or without these functional mutations can then be produced and used as templates for structure-based immunogen design or for the empiric screening of vaccine immunogens. When applied to a new donor, these immunogens may be able to engage germline B cell receptors to elicit favorable variable region mutations that will guide B cell selection toward neutralizing lineages. B cell lineage-targeted vaccine design has been used to guide the development of vaccine candidates that are designed to generate broadly neutralizing antibodies against HIV-1 [28–32] and influenza [33]. However, this approach has not yet been applied to herpes viruses such as CMV, which could be a boon for their vaccine development.

When considering structure-based vaccine design for engaging germline B cell receptors, we are particularly interested in identifying function-enabling somatic hypermutations that may require high-affinity antigenic stimulation. These mutations may occur infrequently during normal affinity maturation due to 1) the location of these mutations in areas of the germline that are rarely targeted by activation-induced deaminase (AID) during somatic hypermutation and/or 2) the requirement of multiple nucleotide mutations within a codon to introduce an observed amino acid substitution [34]. Although the low frequency of these mutations occuring cannot be changed, these functional, "improbable" amino acid mutations could be targeted for high-affinity immunogen stimulation, as identified by structure-based vaccine design.

Based on the low prevalence of gB AD-2S1 antibodies in CMV-seropositive individuals, we hypothesized that improbable AID mutations contribute to the lineage restriction of potently neutralizing antibodies against CMV gB AD-2S1. To address this hypothesis, we examined the lineage maturation of a well-studied, potently neutralizing gB AD-2S1-specific mAb called TRL345 [16, 35]. TRL345 was isolated from single B cell clones from a CMV-seropositive donor with high plasma CMV-neutralizing activity, and based on its potent,

broad neutralization of CMV, this mAb was pursued as a potential passive therapeutic to treat CMV viremia [16, 25, 35]. A Phase 1 clinical trial of TRL345 in healthy volunteers is currently underway, with the goal to develop this as an alternative to the antiviral ganciclovir, which has numerous side effects including neutropenia, nephrotoxicity, and potential muta-genicity that preclude its use for major indications including congenital transmission or the early post-transplant period for hematopoietic cell transplant (ClinicalTrials.gov Identifier NCT05408091).

In this study, we mapped the clonal genealogy of TRL345 from its germline precursor and produced antibodies from its clonal family including inferred ancestral antibodies as well as mature antibodies. Then, using a computational program to estimate the probability of indi-vidual amino acid mutations along maturation pathways, we identified a single improbable heavy chain mutation required for mAb binding and neutralization. This study found that lineage-based vaccine design can be used to identify the sequence and structures of vaccine immunogens that may be capable of eliciting CMV gB AD-2S1-specific potently neutralizing antibodies in an individual.

## Results

### Identification of the TRL345 clonal lineage

We first reconstructed the clonal genealogy of the highly neutralizing gB AD-2S1 mAb TRL345 using a set of previously published mature gB AD-2S1 mAb sequences isolated from the same donor [16, 35]. Using the antibody sequence analysis program Cloanalyst, we grouped the 14 reported mature AD-2S1 mAb sequences into clones. The clone that included mAb TRL345 was comprised of 9 members and utilized a IGHV3-30-3*01 and IGKV3-11*01 pairing, which is consistent with the heavy and light chain V gene segment pairings identified for all previously published gB AD-2S1 mAbs [16, 24, 36–38]. With the 9 TRL345 clonal sequences as input, we then used Cloanalyst to infer the unmutated common ancestor (UCA) sequence representing the B cell receptor of the naïve B cell that gave rise to the TRL345 clone, as well as a maximum likelihood genealogical tree of its clonal lineage (Fig 1).

### High affinity gB AD-2 recognition was acquired early in the TRL345 lineage

After producing the 9 mature mAbs, 8 clonal ancestor intermediates, and UCA mAbs, we measured their binding to gB. The UCA had relatively low binding affinity to both gB AD-2S1 peptide and gB ectodomain (Figs 1, 2A and 2B and S1 Table). However, upon transition from the UCA to the first intermediate (intermediate 8, I8), ELISA binding magnitude for gB AD-2S1 peptide and gB ectodomain increased 151-fold and 37-fold, respectively. Binding affinity for the linear gB AD-2S1, as measured by surface plasmon resonance (SPR), remained unchanged in the UCA to I8 transition, but affinity for gB ectodomain increased 32-fold, to its binding plateau (Figs 1, 2B and 2C and S1 Table). Upon evolution of I8 to the second inter-mediate (I4) in the TRL345 lineage, affinity for the linear gB AD-2S1 increased 5-fold. The high binding magnitude and affinity for both gB AD-2S1 and ectodomain were retained from I4 throughout the rest of the lineage (Figs 1 and 2A–2C and S1 Table). Together, the data indi-cate that high affinity recognition of the AD2 epitope is mediated by the naïve TRL345 B cell and that affinity maturation plateaus by the first or second intermediates along the TRL345 lineage.

Because the gB ectodomain soluble protein may not adequately mimic either the prefusion or postfusion state of gB on the viral envelope or infected cells [17, 39, 40], we measured mAb

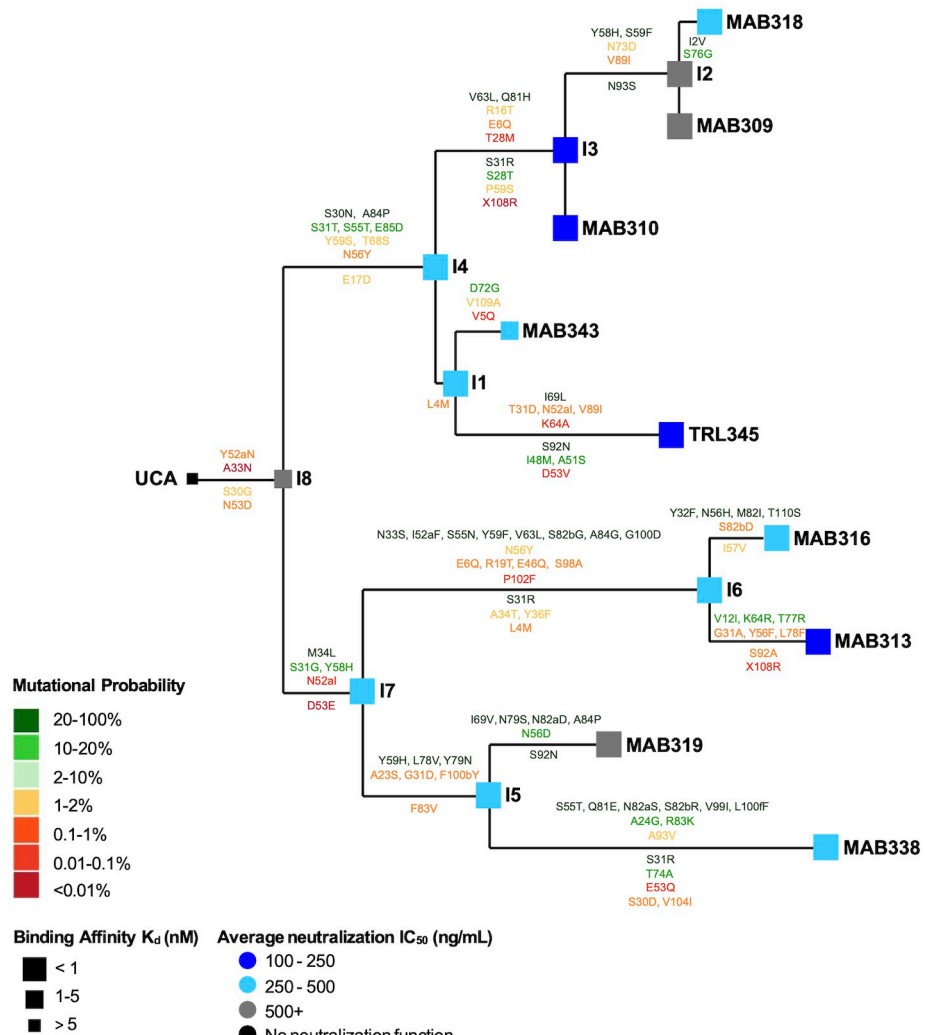

**Fig 1. mAb binding to gB ectodomain and CMV neutralization improved along TRL345 lineage maturation.** We identified 9 clonally related mature mAbs for phylogenetic relatedness, with the phylogenetic tree showing the relatedness using Cloanalyst, and produced clonal lineage mAbs. We measured mAb binding affinity by SPR and neutralization of the Towne, AD169rUL131-GFP, and Toledo CMV strains on MRC-5 fibroblasts. We reported the binding affinity to gB ectodomain as the Kd (nM), calculated as the $k_a/k_d$, from high binding affinity (<1 nM, large box) to moderate (1–5 nM, medium) to low (>5 nM, small). CMV neutralization is reported as the average IC$_{50}$ across all three CMV strains on fibroblasts. The estimated mutations occurring at each transition along lineage maturation are shown above and below the lines of the lineage tree for the heavy and light chains, respectively, and amino acids are numbered according to the Kabat scheme. The probability of a given mutation occurring in the absence of B cell selection was calculated using the ARMADiLLO algorithm, which computationally simulates hypermutation. The mutational probabilities are shown from high probability (>10%, dark green) to moderate (2–10%, light green) to low (<2%, yellow to red text).

binding to full-length gB as expressed on the surface of human epithelial cells, which is a desirable immunogenicity target given that IgG binding to gB expressed on the surface of a cell was identified as an immune correlate of protection against CMV [41]. Of note, gB expression on the surface of a cell may represent a combination of prefusion and postfusion gB forms [17]. Although the UCA had negligible binding to gB AD-2S1 peptide and soluble gB ectodomain, the UCA showed moderate binding to cell-associated gB (AUC of a 3-point dilution series = 97.3 ± 21.2) (Fig 2D and S1 Fig and S1 Table). Binding magnitude for cell-associated

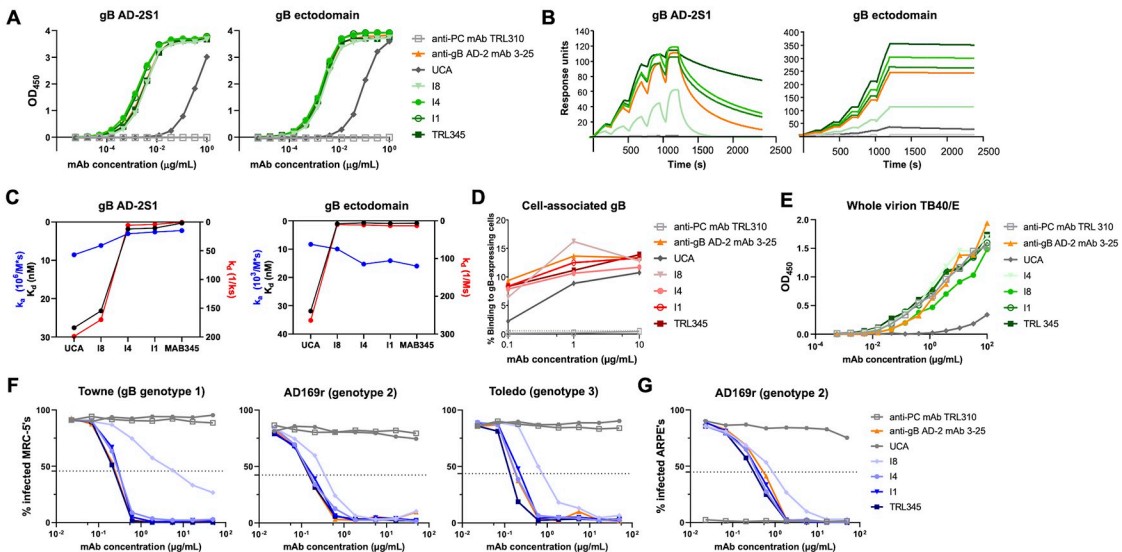

**Fig 2. The TRL345 UCA acquired high binding to gB AD-2S1 and broad CMV neutralization function early in lineage development.** We produced the 17 clonally related mAbs of the TRL345 lineage and quantified antigen binding and neutralization functions. Shown here are the responses of the lineage ancestors of TRL345: (A). Binding magnitude to gB AD-2S1 peptide and gB ectodomain by ELISA. (B). Binding to and avidity for gB AD-2S1 peptide and gB ectodomain by SPR. (C). Binding coefficients for gB AD-2S1 peptide and gB ectodomain by SPR. $K_d$ (nM) was calculated as $k_d/k_a$. (D). Binding to cell-associated gB. Binding of mAbs was determined by coincubating mAbs in a serial dilution with HEK293T epithelial cells-transfected with full-length gB and GFP. The % binding was calculated as the % of GFP-expressing cells bound by the anti-gB AD-2 mAb, detected by flow cytometry. (E). Binding to whole virion CMV strain TB40/E by ELISA. (F). Neutralization of CMV strains Towne, AD169rUL131-GFP (AD169r), and Toledo on MRC-5 fibroblasts. (G). Neutralization of AD169rUL131-GFP on ARPE epithelial cells. Each experiment was run with the negative control anti-CMV pentameric complex (PC) mAb TRL310 and positive control anti-CMV gB AD-2 mAb 3–25. For each figure, data are shown from one experiment, as the mean of two samples run in duplicate. Each figure is representative of results from two or more independent experiments.

gB improved after maturation from UCA to I8 (AUC = 132.6 ± 33.3) then remained stable along the lineage maturation to TRL345 (AUC = 151.6 ± 20.8). Similarly, high binding to whole CMV virions of the TB40/E strain, which expresses gB genotype 1, was acquired in the transition from the UCA to I8 and further refined along lineage maturation (Fig 2E).

## Broad neutralization of CMV was acquired early in the TRL345 lineage

We then quantified the neutralization activity of the TRL345 lineage mAbs against three CMV strains, each of which expressed a different gB genotype. We measured the neutralization of CMV strains Towne (gB genotype 1), AD169 repaired (AD169rUL131-GFP, genotype 2), and Toledo (genotype 3) on MRC-5 fibroblasts and neutralization of AD169rUL131-GFP on ARPE epithelial cells. The UCA did not neutralize any CMV strains on either fibroblasts (Figs 1 and 2F and S1 Table) or epithelial cells (Fig 2G and S1 Table), but it acquired broad neutralization activity in its first lineage branch at I8 (fibroblast neutralization $IC_{50}$ of Towne, AD169rUL131-GFP, and Toledo = 2.78 ± 3.40 μg/mL, 0.83 ± 0.72 μg/mL, and 0.83 ± 0.10 μg/mL, respectively; Epithelial cell $IC_{50}$ of AD169rUL131-GFP = 0.79 ± 0.17 μg/mL). Neutralizing activity further improved along lineage maturation, with 4.8-fold and 2.2-fold increases in average CMV neutralization on fibroblasts and epithelial cells, respectively, after the I8 maturation to I4. The most potently neutralizing antibodies on fibroblasts ($IC_{50}$ < 0.25 μg/mL) were TRL345, MAB310 (I3), and MAB313, and on epithelial cells ($IC_{50}$ < 0.35 μg/mL) were TRL345, MAB343, MAB310 (I3), and MAB313 (Fig 1 and S1 Table).

## Early acquired amino acid mutations were improbable in the absence of B cell selection

We next investigated the probability of each mutation occurring in the absence of B cell selection. During B cell development in germinal centers, B cells undergo somatic hypermutation, wherein AID generates DNA point mutations at immunoglobulin variable regions [42]. B cells with mutations that improve antigen binding avidity undergo subsequent selection. Due to codon degeneracy and biases in AID targeting, certain amino acid changes occur more frequently than others during somatic hypermutation [43]. The acquisition of improbable mutations that contribute to neutralization function can act as rate-limiting steps in the development of neutralizing antibodies [30] and represent high value targets for selection in lineage-based vaccine design strategies [31]. We used the ARMADiLLO computational program to simulate somatic hypermutation and estimate the probability of each observed mutation in the TRL345 clone [30]. Consistent with previous studies, we defined "improbable" mutations as those estimated to occur at <2% frequency in the absence of selection—a frequency corresponding to approximately one mutation per clone per germinal center [30].

We found that all of the mutations acquired by the UCA during its evolution to I8, namely the $V_H$ A33N and Y52aN and $V_L$ S30G and N53D, were improbable in the absence of B cell selection (Fig 1 and S2 Table). Throughout the lineage, of the total 62 heavy chain mutations observed, 7 occurred in CDR1 and 10 in CDR2 regions, with none observed in CDR3. Of the total 29 light chain mutations observed, 6 occurred in CDR1, 1 in CDR2, and 4 in CDR3 (Fig 1). Of the heavy and light chain mutations in the lineage, 37% (23 of 62) and 59% (17 of 29) were estimated to be improbable in the absence of selection, respectively. In TRL345 alone, 6 of 12 total observed heavy chain mutations and 2 of 7 total light chain mutations were improbable (Fig 1 and S2 Table).

## Early, improbable mutations in the UCA were both necessary and sufficient for neutralization

To identify which early mutations conferred neutralizing activity to the UCA, we performed site-directed mutagenesis to either 1) revert the mutations in I8 back to germline or 2) introduce the mutations into the UCA. We produced 14 mutant mAbs with single heavy chain mutations A33N and Y52aN or light chain mutations S30G and N53D or combinations of these mutations, then we quantified mAb binding and neutralization. We found that reversion of single mutations $V_H$ Y52aN, $V_L$ S30G, or $V_L$ N53D in I8 had negligible impact on mAb binding to gB AD-2S1, gB ectodomain, and cell-associated gB (Fig 3A and 3B, S2A and S2B Fig). However, reversion of $V_H$ A33N in I8 decreased binding magnitude to gB AD-2S1, gB ectodomain, and cell-associated gB by 395-fold, 1.7-fold, and 1.4-fold, respectively.

Introduction of $V_H$ A33N to the UCA conferred a 269-fold, 45-fold, and 2.5-fold increase in binding magnitude to the gB AD-2S1 linear epitope, gB ectodomain, and cell-associated gB, respectively (Fig 3A and 3B, S2A and S2B Fig). By contrast, introduction of $V_L$ S30G conferred only 1.8-fold, 13.5-fold, and 1.2-fold increases in binding magnitude to the gB AD-2S1 linear epitope, gB ectodomain, and cell-associated gB, respectively. Introduction of the single mutations $V_H$ Y52aN or $V_L$ N53D conferred 2.0-fold and 8.7-fold increases in binding magnitude to gB ectodomain, respectively, but they had negligible effect on binding to gB AD-2S1 or cell-associated gB. Early, improbable mutations introduced in combination to the UCA demonstrated binding patterns consistent with the single mutants. Thus, mutant mAbs containing $V_H$ A33N had increased binding to gB AD-2S1 linear domain, gB ectodomain, and cell-associated gB, and this pattern was not observed for any other early mutation (Fig 3A and 3B, S2A and S2B Fig).

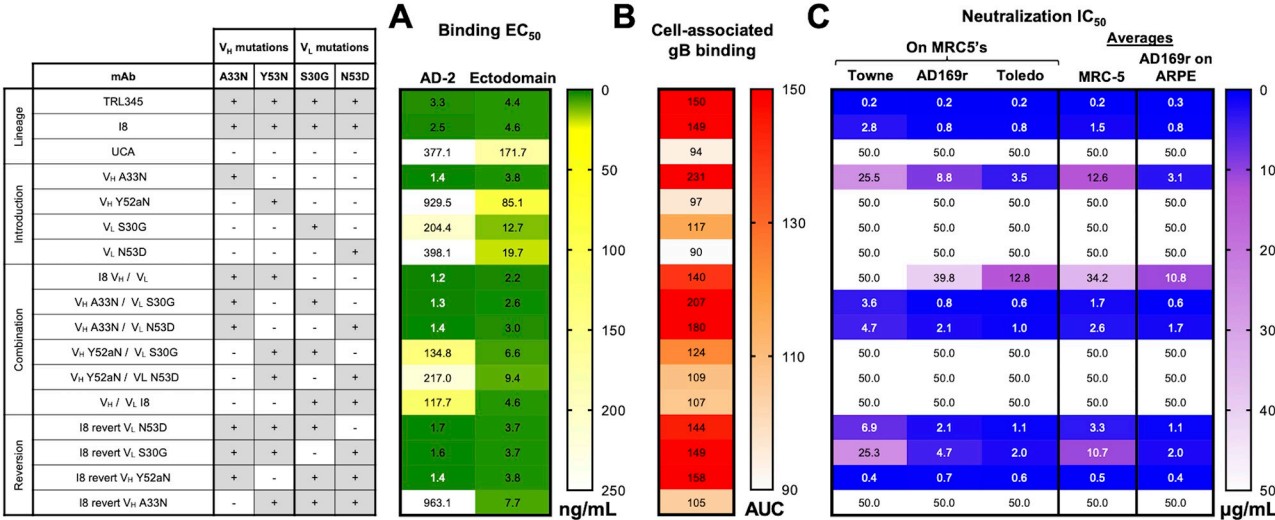

**Fig 3. The $V_H$ A33N mutation was both necessary and sufficient for high gB AD-2 binding, binding to cell-associated gB, and neutralization function.** We produced 14 mutant mAbs wherein we introduced single early, improbable mutations or combinations of mutations to the UCA $V_H$ and $V_L$ chains then measured the following: (A). Binding magnitude of mutant mAbs to gB AD-2S1 peptide and gB ectodomain by ELISA. Data are reported as the mean ELISA $EC_{50}$ (ng/mL) of two or more independent experiments, and the heatmap is colored from high (<50.0 ng/mL, dark green) to moderate (100.0 ng/mL, yellow) to low (>250.0 ng/mL, white) binding. (B). Binding magnitude to cell-associated gB. Binding to HEK293T epithelial cells-transfected with full-length gB was measured in a three-point, 10-fold mAb serial dilution starting at 0.1 μg/mL, to gB-transfected cells and was reported as the area-under-the-curve (AUC). Data are reported as the mean binding AUC of two or more independent experiments, and the heatmap is colored from high (>300 AUC, red) to moderate (200 AUC, orange) to low (<100 AUC, white) binding. (C). Neutralization of CMV strains Towne, AD169rUL131-GFP (AD169r), and Toledo on MRC-5 fibroblasts and AD169r on ARPE epithelial cells. Data are reported as the mean neutralization $IC_{50}$ of two or more independent experiments. The average neutralization on MRC-5's was calculated as the mean neutralization of the average Towne, AD169r, and Toledo neutralization $IC_{50}$'s; the average neutralization on ARPE's was reported as the mean neutralization of the AD169r strain. The heatmap is colored from high (<1.0 μg/mL, blue) to moderate (20.0 μg/mL, purple) to no (>50.0 μg/mL white) neutralization potency.

We then determined the role of these early mutations in mAb neutralizing function. We found that reversion of $V_H$ Y52aN in I8 had negligible impact on neutralization on both fibroblasts and epithelial cells (Fig 3C, S2C and S2D Fig). Reversion of $V_L$ S30G and N53D moderately decreased neutralization of all strains, with the largest impacts on Towne neutralization as observed by decreases in neutralization potency of 9.0-fold and 2.5-fold, respectively. By contrast, reversion of $V_H$ A33N abrogated neutralizing function of all strains on both fibroblasts and epithelial cells. Introduction of the single heavy chain Y52aN or light chain mutations S30G or N53D to the non-neutralizing UCA did not confer neutralization function. Only introduction of the heavy chain A33N mutation conferred neutralizing function, and this was observed for neutralization of all three CMV strains and on both cell types. Addition of $V_L$ S30G or $V_L$ N53D to $V_H$ A33N in the UCA improved neutralizing function on both fibroblasts and epithelial cells, yet addition of $V_H$ Y52aN to $V_H$ A33N decreased mAb neutralizing function (Fig 3). Consistent with the binding patterns of these mutant mAbs, these neutralization results indicate that the improbable heavy chain A33N mutation was both necessary and sufficient for neutralization across multiple CMV strains.

## Heavy chain mutations A33N and A33G confer neutralizing activity in germline mAbs

All human gB AD-2S1-specific mAbs sequenced to date are derived from the IGHV3-30 (or related IGHV3-30-3) and IGKV3-11 germline pairing and are associated with a $V_H$ Gly33, Asn33, or less commonly, Asp33 residue in the mature mAb sequence [44]. We anticipated

that the germline $V_H$ mutations A33N, A33G, or A33D, might represent alternative pathways to achieve neutralization potency.

We first evaluated whether $V_H$ mutations A33G or A33D were also improbable in the absence of B cell selection in their respective lineages. We computationally inferred the UCA's of previously published CMV gB AD-2S1 mAbs 3–25 and ITC88 [27, 45] then simulated somatic hypermutation using the ARMADiLLO algorithm. We found that in 3–25 [45], derived from the IGHV3-30-3*01 allele, the observed mutation $V_H$ A33G occurred at a frequency of 1.94%. In ITC88 [27], derived from the IGHV3-30*04 allele, $V_H$ A33D occurred at a frequency of 1.05%. In combination with the frequency estimates for the mutation $V_H$ A33N in the TRL345 lineage, our data indicated that across multiple donors, functional mutations at the heavy chain residue 33 were statistically improbable, occurring at a frequency of <2%, and likely occurred under pressure of selection.

To determine the functional role of the $V_H$ mutation A33G, we introduced this mutation to the non-neutralizing TRL345 UCA and found that this mutation conferred approximately 15.0-fold and 14.2-fold increases in binding to gB AD-2S1 and gB ectodomain, respectively, with a minimal increase of 1.1-fold in binding to cell-associated gB (Fig 4A and 4B). However, this mutation did not confer CMV neutralizing activity on fibroblasts or epithelial cells (S3A Fig).

We also evaluated the comparative advantages of the $V_H$ A33N and $V_H$ A33G mutations to the UCA of another donor. To identify a UCA with a CDR3 region capable of binding gB AD-2S1, we computationally inferred the UCA from the mAb 3–25 (S4A and S4B Fig). We found that the 3–25 UCA had low binding to gB AD-2S1 (EC50>10 μg/mL) and moderate binding to gB ectodomain (6.1 ± 1.2 μg/mL) and cell-associated gB (22.1 ± 1.8 μg/mL. Fig 4C and 4D). Introduction of $V_H$ A33N to the 3–25 UCA conferred binding to gB AD-2S1 (2.7 ± 2.5 μg/mL) and increased binding magnitude to gB ectodomain and cell-associated gB by 35.2-fold and

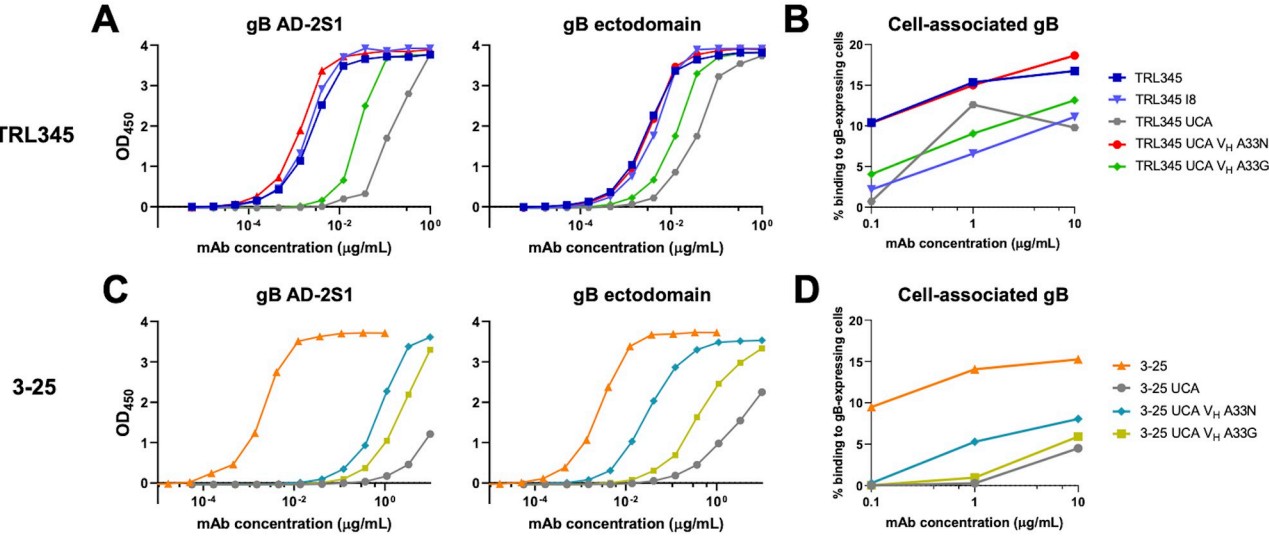

**Fig 4. Introduction of $V_H$ A33N, as compared with $V_H$ A33G, conferred higher binding to gB AD-2S1, gB ectodomain, and cell-associated gB for both the TRL345 UCA and 3–25 UCA antibodies.** We produced 4 mutant mAbs wherein we introduced $V_H$ A33N or A33G mutations, numbered according to the Kabat scheme, to the TRL345 UCA $V_H$ or 3–25 UCA $V_H$ and measured the following: (A). Binding magnitude of TRL345 mutant mAbs to gB AD-2S1 peptide and gB ectodomain by ELISA. (B). Binding of TRL345 mutant mAbs to cell-associated gB. Binding of mAbs was determined by coincubating mAbs in a serial dilution with HEK293T epithelial cells-transfected with full-length gB and GFP. The % binding was calculated as the % of GFP-expressing cells bound by the anti-gB AD-2 mAb, detected by flow cytometry. (C). Binding magnitude of 3–25 mutant mAbs to gB AD-2S1 peptide and gB ectodomain by ELISA. (D). Binding to 3–25 mutant mAbs to cell-associated gB.

4.4-fold, respectively. By contrast, introduction of naturally observed mutation, $V_H$ A33G, to the 3–25 UCA conferred minimal binding, with increases of 1.3-fold, 5.5-fold, and 1.4-fold to gB AD-2S1, gB ectodomain, and cell-associated gB, respectively (Fig 4C and 4D). When introduced into the 3–25 mAb UCA, neither the $V_H$ A33N mutation nor A33G mutation was sufficient to confer CMV neutralizing activity (S4B Fig), suggesting that there may be other mutations required to develop this function for germline pairings different from the IGHV3-30-3*01 and IGKV3-11*01 used in TRL345.

## Glycan shielding in the prevention of gB AD-2S1 antibody binding

In addition to lineage restriction and structural constraints within gB, glycan shielding has been hypothesized to prevent antibody binding to gB AD-2S1 [15, 17, 46]. There are eighteen total N-glycosylation sites within gB, of which three are located in or near the gB AD-2S1 region (amino acids 68–81) at Asn68, Asn73, and Asn95 [17]. We found that mutation of all three gB N-glycosylation sites did not impact TRL345 mAb binding (S5 Fig), suggesting that gB glycosylation is not a major barrier to antibody recognition of gB AD-2S1 by TRL345-like mAb lineages.

## Molecular determinants of gB AD-2S1 linear epitope binding and neutralizing potency

To obtain high-resolution information on the binding of the early intermediate mAb I8 to the gB AD-2S1 linear epitope, we conducted crystallographic studies of TRL345.I8 antigen-binding fragment (Fab) in complex with the linear gB AD-2S1 peptide (65-HRANE-TIYNTTLKYG-79). A crystal in the space group $P2_12_12_1$ with one complex per asymmetric unit diffracted X-rays to a resolution of 1.8 Å. Following molecular replacement and manual building of the model, the structure was refined to an $R_{work}/R_{free}$ of 15.0%/17.4% (S3 Table). This high-resolution structure revealed hydrogen bonding between the side chain of Tyr78 of the gB AD-2S1 peptide and the side chains of $V_H$ Asn33 and Asn52a (both the result of improbable mutations), anchoring the C-terminus of the peptide in a pocket formed by Asn33 of $V_H$ CDR1, Asn52a and Asn57 of $V_H$ CDR2, and the $V_H$ CDR3 loop (Fig 5A and 5B). The side chain of Glu69 of gB AD-2S1 hydrogen bonds with the side chain of Tyr32 of $V_H$ CDR1, anchoring the N-terminus of the peptide, and the side chain of Thr74 of gB AD-2S1 forms hydrogen bonds with the sidechain Ser98 and the mainchain of Val99 of $V_H$ CDR3 (Fig 5B).

We then compared the TRL345.I8 structure with the previously published structure of the potently neutralizing gB AD-2S1-specific mature mAb 3–25 in complex with gB AD-2S1 (PDB ID: 6UOE), which also uses the IGHV3-30-3*01 and IGKV3-11 pairing, and the mature gB AD-2S1-specific mAbs 8F9 (PDB ID: 3EYF) and KE5 (PDB ID: 4HHA), which both contain a Gly33 at the $V_H$ allele [17, 24, 38, 40, 47, 48]. The structures of the peptides in all four complexes are highly similar, with an RMSD of 0.9 Å for 11 Cα atoms for TRL345.I8 and 3–25, 0.9 Å for 10 Cα atoms for TRL345.I8 and 8F9, and 0.4 Å for 10 Cα atoms for TRL345.I8 and KE5, despite numerous amino acid differences in the heavy chains at the binding interface (Fig 5B–5H, S4A and S4B Fig). The structural similarity of the peptide suggests that this particular conformation of the gB AD-2S1 epitope may represent the conformation adopted in the prefusion form of gB.

Interactions between gB AD-2S1 and the $V_L$ of 3–25 are recapitulated in the TRL345.I8 complex, with the side chain and main chain of Asn73 of gB AD-2S1 hydrogen bonding with the side chains of Tyr32 and Arg91 of the $V_L$ CDR1 and CDR3, respectively, and the side chain of gB T75 hydrogen bonding with the side chain of $V_L$ CDR3 Trp94 (Fig 5D). Notably, these interactions recapitulated in both the TRL345.I8 and 3–25 complexes occur at shared

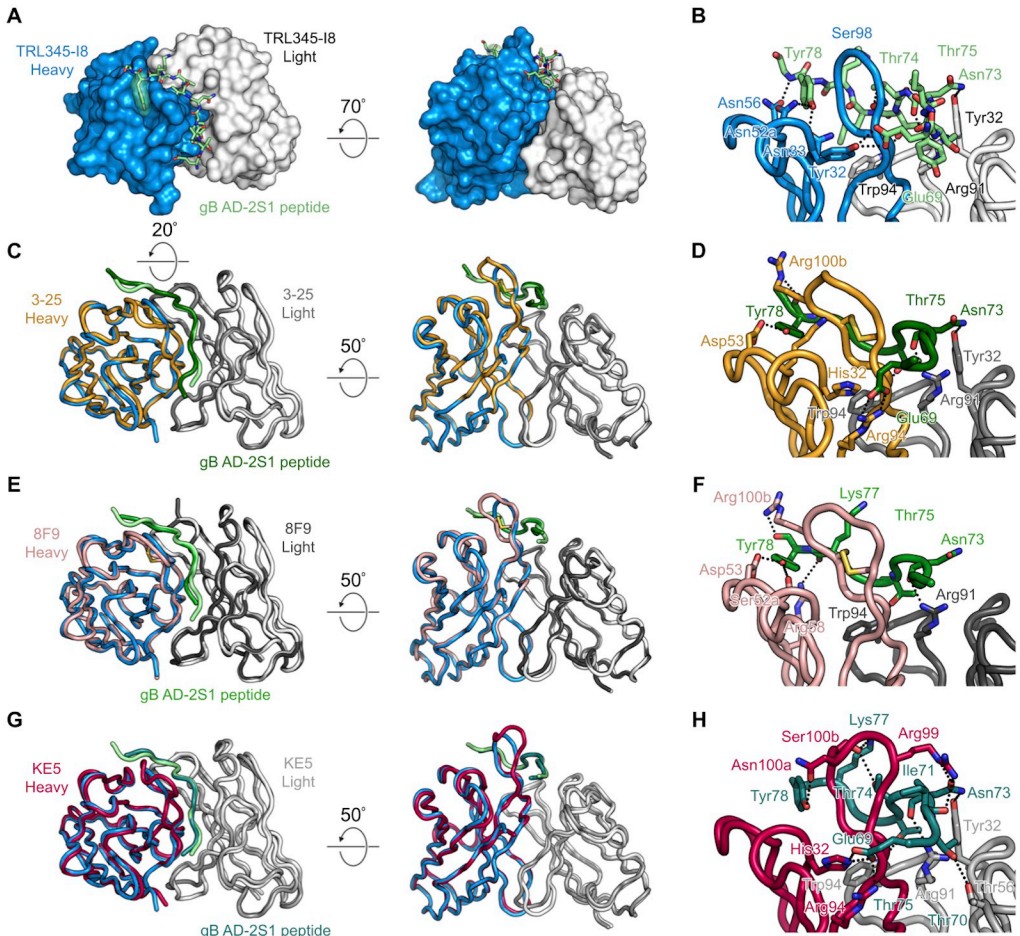

**Fig 5. Crystal structure of TRL345-I8 Fab bound to gB AD-2S1 peptide.** (A). The TRL345-I8 Fab is shown as a molecular surface, with the heavy chain in blue and light chain in white. The gB AD-2S1 peptide is shown as light green sticks, with residues that form hydrophobic contacts with the TRL345-I8 Fab shown as transparent light green surfaces. (B). The TRL345-I8 Fab interface with the gB AD-2S1 peptide. (C). Superposition of TRL345-I8 Fab and 3–25 Fab (PDB ID: 6UOE) AD-2S1 complexes. (D). The 3–25 Fab interface with the gB AD-2S1 peptide. (E). Superposition of TRL345-I8 Fab and 8F9 Fab (PDB ID: 3EYF) AD-2S1 complexes. (F). The 8F9 Fab interface with the gB AD-2S1 peptide. (G). Superposition of TRL345-I8 Fab and KE5 Fab (PDB ID: 4HHA) AD-2S1 complexes. (H). The KE5 Fab interface with the gB AD-2S1 peptide. In all panels, the TRL345-I8 heavy and light chains are colored blue and white, respectively, and the AD-2S1 peptide to which it is bound is colored light green. In panels B, D, F, and H, Fabs and bound peptides are shown as ribbons, with interacting residues shown as sticks. Hydrogen bonds and salt bridges are shown as black dotted lines. Oxygen atoms are red, nitrogen atoms are blue, and sulfur atoms are yellow. Numbering is in the Kabat scheme.

germline-encoded amino acids (Fig 1, S4A and S4B Fig). The $V_L$ interactions in both the TRL345.I8 and 3–25 complexes are largely shared by the 8F9 and KE5 complexes (Fig 5B–5H). However, the side chain of 8F9 $V_L$ CDR1 Tyr32 doesn't interact with the side chain of N73 of gB AD-2S1 (Fig 5E and 5F), and the side chain of KE5 $V_L$ CDR2 Thr56 makes an additional hydrogen bond with the side chain of Thr70 of gB AD-2S1 (Fig 5G and 5H).

In all four complexes, the main chain atoms of gB AD-2S1 Thr75, Leu76, and Tyr78 and the side chain of Thr75 form hydrogen bonds with main chain atoms of the $V_H$ CDR3. However, whereas the side chain of gB AD-2S1 Thr74 hydrogen bonds with the side chain TRL345.I8 $V_H$ CDR3 S98, the side chain of Thr74 hydrogen bonds with the main chain of Cys98 of the 3–25 $V_H$ CDR3, the side chain of which forms a disulfide bond with Cys100B, stabilizing the

longer 3–25 mAb CDR3 loop (the 3–25 $V_H$ CDR3 is one residue longer than TRL345.I8 $V_H$ CDR3, Fig 5C and 5D). Although the Thr74 AD-2S1 interaction with VH CDR3 Cys98 is absent in 8F9, the sidechain of Cys98 also forms a stabilizing disulfide bond with Cys100c (Fig 5E and 5F). In contrast, the sidechain of Thr74 hydrogen bonds with the sidechain of Ser100b of the KE5 $V_H$ CDR3, which is not stabilized by a disulfide bond despite its longer CDR3 length compared to I8 (Fig 5G and 5H).

In assessing both the general similarity and key differences in the interactions between the early TRL345 intermediate and mature AD-2S1-specific neutralizing mAbs with gB AD-2S1, we have identified a structural basis for the increase in binding affinity following the transition from the UCA of TRL345 to I8 that is present across a number of mature mAbs with similar function to TRL345, suggesting that there is a common AD2-S1 epitope confirmation in engaging precursor mAbs that can develop into potently neutralizing responses against CMV.

## Discussion

Advancements in the antibody-to-vaccine approach have successfully enabled the translation of antibody lineages and structures to effective vaccine immunogens. Recently, structure-based design was used to engineer a candidate vaccine against RSV by stabilizing the prefusion F conformation and preserving neutralization-sensitive epitopes on the vaccine antigen [49, 50]. For CMV, structure-based vaccine design of CMV gB to elicit potent neutralization has faced challenges, partly due to the high flexibility of the gB AD-2S1 region, a target of broadly and potently neutralizing antibodies. Previous structural studies of the full-length gB protein, in both the prefusion and postfusion conformations, have not been able to resolve the gB AD-2S1 conformation [17, 21, 38, 40, 47]. The recently solved prefusion structure, in which AD-2 remained unresolved, suggested that the N-terminal flexible region containing gB AD-2 may bind gH/gL in the pentameric complex to initiate cell fusion [21]. In our study, we solved the atomic structure of a gB AD-2S1 peptide bound by an early B cell precursor mAb with high affinity. Future studies should aim to stabilize the gB AD-2S1 peptide in this conformation and assess whether it is capable of eliciting potently neutralizing antibodies.

Our study was the first to reveal that only a low level of mAb affinity maturation, specifically a single mutation in a single donor, is required for antibody precursors to achieve strong epitope binding and CMV neutralization. A previous study of the gB AD-2S1 mAb ITC88, derived from the IGHV3-30*04 and IGKV3-11 pairing, determined that induction of the $V_H$ A33D mutation was critical for high-affinity binding [27]. Consistent with these results, we found that the $V_H$ A33N mutation in the TRL345 lineage improved binding to gB AD-2S1 peptide and gB ectodomain not only for the TRL345 UCA but also for the UCA of a second donor (Figs 1, 2 and 4). This mutation was also sufficient to confer potent and broad neutralizing function to the non-neutralizing TRL345 UCA, with further improvements in neutralizing function with the addition of the $V_L$ S30G and N53D mutations (Fig 3). By comparison, broadly neutralizing antibodies against HIV-1 require multiple mutations and can lose neutralizing function along lineage maturation, posing significant challenges for mutation-guided vaccine design strategies [51]. Thus, a B cell lineage-based, mutation-guided vaccine approach to the design, antigenicity screening, and immunogenicity metrics of gB AD-2S1 immunogens may be feasible.

We identified the structural basis of affinity maturation from the TRL345 UCA to early intermediate mAbs. In particular, the $V_H$ A33N and Y52aN mutations enabled hydrogen bonding to the side chain of Tyr78 of the gB AD-2S1 peptide, anchoring the C-terminus of the peptide in a pocket (Fig 5B). Notably, the mature TRL345 mAb contains an N52aI substitution, indicating that there may be key differences between the structures of early lineage and

mature mAbs in binding to the flexible peptide. Moreover, there were several differences in the interactions between gB AD-2S1 and TRL345.I8 and the mature mAbs 3–25, 8F9, and KE5, suggesting that there may be multiple pathways to developing potently neutralizing responses against CMV.

This study has several limitations. Due to the limited number of anti-gB AD-2S1 B cell clones sequenced to date, we were only able to investigate the anti-gB AD-2S1 B cell lineage of a single donor. Accordingly, we were not able to observe whether there may be parallel evolution of key mutations in other donors, and follow-up studies are needed to determine whether other mutations in addition to or instead of the $V_H$ A33N mutation can confer gB AD-2S1 neutralization function to germline mAbs that share genetic and structural characteristics across donors. We anticipate that future studies may identify other key variable region mutations involved in CMV neutralization that would also be desirable for gB immunogens to engage and elicit. Moreover, the TRL345 antibody used the IGHV3-30-3*01 allele, but there are multiple highly similar $V_H$ genes, such as the IGHV3-30 and IGHV3-30-3, that might contribute to these responses and that might be present in other individuals. Similarly, there may be conserved light chain mutations in other related alleles. Future studies should pursue a more thorough comprehensive mutational mapping of representatives of other CMV gB AD-2S1 mAb lineages that utilize other alleles.

There are additional challenges facing this approach for vaccine design. This study did not address the level of antigen affinity required to target germline B cell precursors *in vivo*. Follow-up studies should be performed to create gB antigens with high binding affinity to B cell precursors of gB AD-2S1-binding mAbs, such as those gB immunogens identified here, and test them in animal models for their ability to elicit gB AD-2S1-specific antibodies. Ideally, these studies would be performed in models with human gene knock-ins that are representative of the germline repertoire of human B cells. It is possible that the gB immunogens identified via this strategy have high affinity to B cell precursors but are unable to initiate neutralizing lineages. Furthermore, there may be low inherent frequency of these B cell precursors across donors, which should be assessed in unbiased B cell repertoire sequencing studies. Additionally, the current versions of algorithms used in this study do not calculate the probability of insertions or deletions in the antibody clonal lineages, which may identify additional targets, and this vaccine strategy is limited in its ability to elicit these types of mutations.

This study determined that only a low level of affinity maturation, even a single mutation, is required for the generation of potently neutralizing CMV gB AD-2S1-specific mAbs and suggested the feasibility of B cell lineage-based vaccine design for gB AD-2S1 immunogens. Our combined approach of B cell receptor sequence analyses, computational modeling, functional assessments, and x-ray crystallography of CMV gB AD-2S1 mAbs enabled the identification of gB AD-2S1 peptide structure which may be capable of eliciting neutralizing antibodies by engaging non-neutralizing early lineage precursors. This lineage-based strategy could be further applied to define CMV gB AD-2S1 peptide structures that can engage gB AD-2S1-specific early antibody intermediates, enhancing the design of gB immunogens to target the induction of potent CMV-neutralizing responses that may be required for effective CMV vaccination.

## Materials and methods

### Study design

The objectives of this study were to assess whether potently neutralizing antibodies against CMV gB AD-2S1 are lineage-restricted through the requirement of improbable AID mutations for binding and neutralizing functions. To address this hypothesis, we evaluated the

following experimental units: 18 mAbs in the clonal genealogy of the neutralizing gB AD-2S1 mAb TRL345 [16, 35] including the TRL345.UCA, 14 mutated TRL345.UCA mAbs with single or combinations of early lineage mutations, the potently neutralizing mAb 3–25 [45], the 3-25. UCA, and 2 mutated 3-25.UCA mAbs. In each experiment, each sample was run in duplicate, and each experiment was performed two or more times.

## Peptide and protein production

The gB AD-2S1 peptide was synthesized by ThermoFisher as the sequence HRANE-TIYNTTLKYG, which includes the underlined, minimal gB AD-2S1 epitope. The gB ectodomain protein was produced in-house by transfection of Expi293i cells and lectin purification. Mutant gB ectodomain proteins with one or more of the following mutations: Asn68Q, Asn73Q, Asn95Q, were produced in-house. Site-directed mutagenesis was performed on the gB ectodomain plasmid using the QuikChange Lightning Multi Site-Directed Mutagenesis Kit (Agilent Technologies), then mutations were confirmed by Sanger sequencing.

## Production of lineage mAbs and mutant mAbs

Antibody genes were synthesized by Genscript and recombinantly produced in a human IgG backbone. Single amino acid mutations were introduced or reverted by site-directed mutagenesis, using the QuikChange Lightning Multi Site-Directed Mutagenesis Kit (Agilent) according to manufacturer's protocol. Sequence efficiency was confirmed by Sanger sequencing.

## mAb binding to soluble peptides and proteins by ELISA

mAb binding to gB AD-2S1 peptide and gB ectodomain were measured by 384-well plate ELISA. Plates were coated overnight at 4˚C with 45 ng gB AD-2S1 or gB ectodomain per well then blocked in assay diluent (1× PBS pH 7.4 containing 4% whey, 15% normal goat serum, and 0.5% Tween-20). mAbs were plated in a 12-point 3-fold serial dilution at a starting concentration of 1 μg/mL (1 μg/mL to $5.7^*10^{-6}$ μg/mL), in duplicate. Binding was detected by goat-anti human HRP-conjugated IgG secondary (Jackson ImmunoResearch). Plates were developed using the SureBlue Reserve tetramethylbenzidine (TMB) peroxidase substrate (KPL). Data are reported as the half-maximal effective concentration ($EC_{50}$). Each mAb was run in two or more independent experiments.

## mAb binding to soluble peptides and proteins by surface plasmon resonance (SPR)

The kinetics and affinity of the binding interactions between monoclonal antibodies and the CMV gB ectodomain protein and the CMV gB AD-2S1 peptide were assessed by surface plasmon resonance (SPR) on a Biacore T200 platform (Cytiva) at 25˚C in HBS-EP+ (10 mM HEPES, 150 mM NaCl, 3 mM EDTA, 0.05% v/v Surfactant P20, pH 7.4) running buffer. Monoclonal antibodies (10 μg/mL) were non-covalently captured on the surface of a Series S Sensor Chip Protein A (Cytiva) by injection for 60 seconds at a flow rate of 5 μL/min. Single-cycle kinetic titration analyses were performed by sequential 180 second injections of 5 antigen concentrations, followed by a 1200 second dissociation phase, at 30 μL/min. The CMV gB ectodomain protein was assayed in a two-fold dilution series from 1.25–20 μg/mL, and the CMV gB AD-2S1 peptide was assayed in a two-fold dilution series from 6.25–100 ng/mL. The Protein A surface was regenerated after each antigen injection with a 30 second injection of 10 mM glycine-HCl, pH 2.0, at a flow rate of 30 μL/min, allowing for subsequent analysis of all mAb-antigen pairs. Binding data was analyzed using the Biacore T200 Evaluation software

(v2.0, Cytiva). Binding profiles were reference subtracted using a negative control Synagis mAb (anti-flu) surface and running buffer injections. Curve fitting analysis was performed using either a 1:1 Langmuir model or a heterogeneous ligand model. For interactions fit with the heterogeneous ligand model, affinities were calculated using the fast kinetic components of the model fit. Some interactions reported dissociation rate constants ($k_d$) beyond the limit of detection of the software ($< 1.00 \times 10^{-5}$ s$^{-1}$). In such cases, the values were recorded as $1.00 \times 10^{-5}$ and affinity calculations are reported as the upper limit of the respective value.

## mAb binding to cell-associated gB

HEK293T cells at 50% confluency in a T75 flask were cotransfected using the Effectine Transfection Reagent (Qiagen) with DNA plasmids expressing GFP (gift of Maria Blasi, Duke University) with or without plasmids expressing the full-length gB ORF from the autologous Towne strain (SinoBiological). After incubation for 2 days at 37°C and 5% $CO_2$, transfected cells were washed with Dulbecco's PBS (DPBS) pH 7.4 (Gibco) then removed from the flask by gently rinsing with Trypsin-EDTA 0.05% with phenol red (ThermoFisher). Cells were resuspended in wash buffer [DPBS pH 7.4 + 1% fetal bovine serum (FBS)] then manually enumerated for count and viability using trypan blue (ThermoFisher). Cells were plated in 96-well V-bottom plates (Corning) at 100,000 live cells/well, then centrifuged at 1200 $g$ for 5 minutes. Supernatant was discarded. Cells were co-incubated with gB mAbs at a 3-point, 10x-fold serial dilution starting at 10 μg/mL (10 μg/mL to 0.1 μg/mL), in duplicate for 2 hours at 37°C and 5% $CO_2$. Cells were washed and resuspended in live/dead Near-IR or Aqua cell stain (ThermoFisher) diluted to 1:1000 for 20 minutes incubation at room temperature. Cells were washed then coincubated with PE-conjugated mouse anti-human IgG Fc (Southern Biotech) diluted to 1:200 for 30 minutes at 4°C. Cells were washed twice and fixed with 1% formalin for 15 minutes. Cells were washed twice then resuspended in PBS pH 7.4. Events were immediately acquired on an LSR II (BD Biosciences) using a high-throughput sampler (HTS). The threshold for PE positivity was defined as 99% of the PE binding by the anti-CMV pentameric complex mAb TRL310 at 10 μg/mL. The % of PE-positive cells was calculated from the live, GFP-positive cell population and reported as the average for each sample run in duplicate. Each sample was run in two or more independent experiments.

## mAb binding to whole virions

384-well ELISA plates were coated with 33 PFU/well TB40/E diluted in 0.1 M sodium bicarbonate buffer then incubated overnight before blocking. mAbs were plated in a 12-point 3-fold serial dilution at a starting concentration of 100 μg/mL (100 μg/mL to $5.7^*10^{-4}$ μg/mL), in duplicate. Binding was detected by goat-anti human HRP-conjugated IgG secondary (Jackson ImmunoResearch). Plates were developed using the SureBlue Reserve tetramethylbenzidine (TMB) peroxidase substrate (KPL). Data are reported as the half-maximal effective concentration ($EC_{50}$). Each mAb was run in two or more independent experiments.

## Neutralization

Neutralization was measured by high-throughput Cellomics bioimaging and fluorescence as previously described [52]. In brief, MRC-5 fibroblasts were seeded in 384-well clear, flat-bottom plates then incubated at 37°C and 5% $CO_2$ until ~90% confluent. Human CMV virus strains Towne (gB genotype 1), BadrUL131-GFP (genotype 2), or Toledo (genotype 3) at an MOI = 2 were coincubated with mAbs in an 8-point, 3-fold serial dilution in cell media (RPMI + 10% FBS) for 2 hours at 37°C. All mAbs were coincubated at a starting concentration of 50 μg/mL (50 μg/mL to 22.9 ng/mL), except the 3–25 mutant mAbs which were coincubated at

a starting concentration of 500 μg/mL (500 μg/mL to 228.6 ng/mL). Virus and antibody mixtures were added to cells and allowed to incubate for 16 to 24 hours at 37˚C and 5% $CO_2$. Then, cells were fixed in 3.7% formaldehyde for 10 minutes at room temperature. Plates with Towne virus were stained with mouse anti-human CMV IE1 (MAB810, Millipore) then goat anti-mouse IgG-AF488 (Millipore). Nuclear staining was performed in all plates by DAPI (ThermoFischer Scientific). Plates were imaged using a Cellomics CellInsight CX5 fluorescent reader, and the number of total cells and infected cells was enumerated by the number of cells expressing DAPI and GFP, respectively. The 50% inhibitory concentration ($IC_{50}$) of each mAb was calculated according to the Reed and Muench method, wherein the $IC_{50}$ was defined as the mAb concentration at which there was 50% maximal infection, based on wells containing cells and virus only. This calculation was performed in GraphPad Prism version 9.0 using the non-linear regression 4-point sigmoidal function. Each sample was run in two or more independent experiments.

## Crystallization and structure determination

The I8 Fab was generated by digesting I8 IgG at 1 mg/mL in PBS pH 7.4 with Lys-C protease at a ratio of 1:4000 (w/w) overnight at 37 ˚C. The reaction was quenched with a Roche EDTA protein inhibitor tablet (Sigma-Aldrich) at 1X concentration and the solution was then passed over a CaptureSelect IgG-CH1 column (ThermoFischer Scientific) to separate the Fab fragment from the Fc portion. The elution was concentrated in a 10 kDa molecular weight cutoff Amicon Ultra Centrifugal filter (Millipore Sigma) and purified by size-exclusion chromatography with a Superdex 200 Increase 10/300 column (GE Healthcare) in 2 mM Tris-Cl pH 8.0, 200 mM NaCl, 0.02% $NaN_3$ buffer. Purified I8 Fab was concentrated to 12.0 mg/mL in the aforementioned buffer, mixed with a 2.5-fold molar excess of 1.0 mg/mL gB AD-2S1 peptide (65-HRANETIYNTTLKYG-79) dissolved in DMSO, and incubated for 30 minutes at 4 ˚C. Crystallization screens were then performed using sitting-drop vapor diffusion with 200 nL drop volumes in either 1:1 or 1:2 protein:reservoir mixtures. Within a few days, diffraction-quality crystals formed in a drop composed of a 1:1 ratio of protein:reservoir solution containing 0.2 M magnesium chloride hexahydrate, 25% (w/v) PEG 3350, and 0.1 M Bis-Tris pH 5.5. Crystals were soaked in reservoir solution supplemented with 20% (v/v) glycerol before they were plunge frozen in liquid nitrogen. X-ray diffraction data were collected remotely at the 19ID beamline (Advanced Photon Source, Argonne National Laboratory). Data were indexed and integrated in iMOSFLM (45) and then merged and scaled to a resolution of 1.80 Å using AIMLESS (46). Conventional Kabat numbering was used consistent with previous structural studies [53–55] such that future structural studies can refer to identified mutations of interest. A Fab homology model generated from PDB IDs 6ZFO and 6UOE and the peptide models from PDB 6UOE, 3EYF, and 4HHA were used with PHASER (47) to find a molecular replacement solution. The structure was then iteratively refined in PHENIX (48) and manually built in Coot (49). All crystallographic software programs used in this project were compiled and configured by SBGrid (50).

## Antibody sequence analysis

The clonal membership of TRL345 was determined by partitioning the sequences of 14 mature anti-gB AD-2S1 mAbs (Patent # US10,030,069B2) into clones using Cloanalyst, version 2007 (https://www.bu.edu/computationalimmunology/cloanalyst/). With the 9 observed mature TRL345 sequences as input, we also used Cloanalyst to reconstruct the TRL345 genealogical tree which included inference of the TRL345 UCA sequence (PMID:24555054). To estimate the probability of antibody mutations prior to antigenic selection, we used the ARMADiLLO

program (https://armadillo.dhvi.duke.edu.), which computationally simulates somatic hypermutation [30]. We defined improbable mutations as those occurring with <2% probability, as previously described [30].

## Software

Phylogenetic analysis was performed using Cloanalyst version 7. Mutation analysis was performed using ARMADiLLO. Statistical tests were performed in GraphPad Prism version 9.0. Figures were created in GraphPad Prism version 9.0. Fig 5 was created in PyMOL version 2.4.1 (Schödinger, LLC).

## Statistical analyses

Statistical tests were performed as described in the figure legends, where applicable. Nonlinear regression curve fitting was performed to calculate the $EC_{50}$ and $IC_{50}$ values. The relative contribution of individual amino acid mutations to mAb neutralization function was calculated by linear regression. Statistical significance was calculated using a nonparametric two-tailed Wilcoxon matched-pairs signed-rank test. All tests were performed in GraphPad Prism version 9.0.

## Supporting information

**S1 Fig. Gating strategy to determine the % of binding to cell-associated gB.**
(PDF)

**S2 Fig. The VH A33N mutation was both necessary and sufficient for high gB AD-2 binding, binding to cell-associated gB, and neutralization function.**
(PDF)

**S3 Fig. Introduction of the VH A33G mutation to the UCA of either the TRL345 or 3–25 lineages did not confer neutralizing function.**
(PDF)

**S4 Fig. Sequence alignment of anti-gB AD-2S1 mAbs from the TRL345 and 3–25 lineages.**
(PDF)

**S5 Fig. Binding of TRL345 to gB ectodomain with and without mutations at N-glycosylation sites.**
(PDF)

**S1 Table. Compiled binding and neutralization responses for TRL345 lineage mAbs and 3–25 mature mAb.**
(PDF)

**S2 Table. Mutational probability analysis using ARMADiLLO algorithm.**
(PDF)

**S3 Table. X-ray crystallographic data collection and refinement statistics.**
(PDF)

## Acknowledgments

We would like to acknowledge our many wonderful collaborators for their support. In particular, we would like to thank Zhiqiang An and Xiaohua Ye for providing plasmids and antibodies and Tong-Ming Fu, Dai Wang, Zhiqiang An, Micah Luftig, and Rory Henderson for their

consultation and guidance. Results shown in this report are derived from work performed at Argonne National Laboratory, Structural Biology Center (SBC) at the Advanced Photon Source. SBC-CAT is operated by UChicago Argonne, LLC, for the U.S. Department of Energy, Office of Biological and Environmental Research under contract DE-AC02-06CH11357.

## Author Contributions

**Conceptualization:** Jennifer A. Jenks, Jason S. McLellan, Kevin Wiehe, Sallie R. Permar.

**Data curation:** Jennifer A. Jenks, Madeline R. Sponholtz, Jason S. McLellan, Kevin Wiehe, Sallie R. Permar.

**Formal analysis:** Jennifer A. Jenks, Madeline R. Sponholtz, Amit Kumar, Daniel Wrapp, Sravani Venkatayogi, Krithika Karthigeyan, Megan Connors, Jason S. McLellan, Kevin Wiehe, Sallie R. Permar.

**Funding acquisition:** Jennifer A. Jenks, Jason S. McLellan, Kevin Wiehe, Sallie R. Permar.

**Investigation:** Jennifer A. Jenks, Sharmi Amin, Madeline R. Sponholtz, Amit Kumar, Daniel Wrapp, Krithika Karthigeyan, Megan Connors, Melissa J. Harnois, Bhavna Hora, Eric Rochat, Jason S. McLellan, Kevin Wiehe, Sallie R. Permar.

**Methodology:** Jennifer A. Jenks, Madeline R. Sponholtz, Kevin Wiehe, Sallie R. Permar.

**Project administration:** Jennifer A. Jenks, Daniel Wrapp, Jason S. McLellan, Kevin Wiehe, Sallie R. Permar.

**Resources:** Amit Kumar, Joshua J. Tu, Sarah M. Valencia, Eric Rochat, Jason S. McLellan, Kevin Wiehe, Sallie R. Permar.

**Software:** Sravani Venkatayogi, Kevin Wiehe.

**Supervision:** Amit Kumar, Sarah M. Valencia, Jason S. McLellan, Kevin Wiehe, Sallie R. Permar.

**Validation:** Jennifer A. Jenks, Sharmi Amin, Madeline R. Sponholtz, Kevin Wiehe, Sallie R. Permar.

**Visualization:** Jennifer A. Jenks, Sharmi Amin, Madeline R. Sponholtz, Sravani Venkatayogi, Jason S. McLellan, Kevin Wiehe, Sallie R. Permar.

**Writing – original draft:** Jennifer A. Jenks, Madeline R. Sponholtz, Jason S. McLellan, Kevin Wiehe, Sallie R. Permar.

**Writing – review & editing:** Jennifer A. Jenks, Madeline R. Sponholtz, Amit Kumar, Jason S. McLellan, Kevin Wiehe, Sallie R. Permar.

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
