## [Decision Letter · Decision Letter 0]

12 Sep 2022

Dear Dr. Permar,

Thank you very much for submitting your manuscript "A single, improbable B cell receptor mutation confers potent neutralization against cytomegalovirus" for consideration at PLOS Pathogens. As with all papers reviewed by the journal, your manuscript was reviewed by members of the editorial board and by several independent reviewers. The reviewers appreciated the attention to an important topic. Based on the reviews, we are likely to accept this manuscript for publication, providing that you modify the manuscript according to the review recommendations.

The reviewers were generally favorable to your provocative manuscript. However, one reviewer notes that your study does not address antibody engagement of gB in a physiologic context, e.g., in a virion and with a "clinical" virus strain. I agree that this is an important issue, and that such data would improve the manuscript. Whether this is best shown as two individual experiments (virion binding and clinical strain neutralization) or a single experiment (clinical strain virion binding) is debatable. The reviewers also ask for some more cautious interpretations and explicit details of the model and its future use, which I also think are reasonable requests.

Sincerely,

Robert F. Kalejta

Associate Editor

PLOS Pathogens

Blossom Damania

Section Editor

PLOS Pathogens

Kasturi Haldar

Editor-in-Chief

PLOS Pathogens

orcid.org/0000-0001-5065-158X

Michael Malim

Editor-in-Chief

PLOS Pathogens

orcid.org/0000-0002-7699-2064

The reviewers were generally favorable to your provocative manuscript. However, one reviewer notes that your study does not address antibody engagement of gB in a physiologic context, e.g., in a virion and with a "clinical" virus strain. I agree that this is an important issue, and that such data would improve the manuscript. Whether this is best shown as two individual experiments (virion binding and clinical strain neutralization) or a single experiment (clinical strain virion binding) is debatable. The reviewers also ask for some more cautious interpretations and explicit details of the model and its future use, which I also think are reasonable requests.

Reviewer Comments (if any, and for reference):

Reviewer's Responses to Questions

**Part I - Summary**

Reviewer #1: In this study, Jenks et al. investigated the genealogy of B cells encoding potently neutralizing anti-gB AD-2S1 antibodies from a single patient. By computational analysis, they inferred an unmutated common ancestor (UCA). Along this prediction, they characterized the binding and function of early lineage ancestors and found that a single amino acid V(H) mutation A33N conferred broad neutralizing activity to the non-neutralizing UCA antibody. The A33N was predicted to be an improbable mutation rarely generated by somatic hypermutation. Crystallographic studies indicated that this mutation improved critical contacts with the gB AD-2S1 epitope.

Overall, the study is well designed and clearly presented. The data are convincing and generally support the conclusions (one important exception is mentioned below). The study provides a plausible explanation for the fact that neutralizing antibodies directed against AD-2S1 are found rather infrequently in HCMV-infected individuals and not at all in individuals vaccinated with a gB vaccine. This information might indeed be important for rational vaccine design. One important limitation of the study is the fact that the antibody genealogy was analyzed in a single patient. Hence it is impossible to decide whether the conclusions drawn from this patient’s antibodies are widely applicable or specific for a subset of the human population. The authors state repeatedly that their study demonstrates the feasibility lineage-based vaccine design to target induction of CMV gB AD-2S1-specific potently neutralizing antibodies. This is an overstatement. They did not show that B cell lineage selection in a vaccinated individual can actually be improved using the data from the present study.

Reviewer #2: This is an interesting study that addresses an important immunological question concerning the maturation of antibody responses against a key neutralising epitope in gB - AD2.

In general, the project is performed well and presented in a logical manner. although based on a single donor this is mitigated by the well established highly restricted VDJ usage for AD2.

In this reviewer's opinion there are not major weaknesses although there does appear to be obvious experiments missing which should temper some of the interpretations made. This can be solved with the addition of some additional simple experiments and/or more refined interpretations of the data.

Reviewer #3: In this interesting paper by Jenks and colleagues, novel observations are reported that have potentially a high impact on the as-yet elusive goal to develop a CMV vaccine. Since CMV is a leading cause of infant hearing loss and developmental disability, there is a great interest (particularly among pediatric infectious diseases specialists) in developing a CMV vaccine. It has been presumed that neutralizing antibodies elicited by vaccination will be key in protection, although the senior (corresponding) author of this paper has been a leader in elucidating the role of non-neutralizing antibodies in protection against CMV infection, particularly in the context of subunit gB vaccination using the first-generation gB/MF59 vaccine.

These studies extend previous work about gB vaccine. Of longstanding interest has been the elucidation of the importance of antibody against key epitopes in the gB molecule, since this can inform and direct subunit vaccine design. The AD-2 epitope is of unique interest, because of the fact that anti-AD2 responses are, and because AD-2 responses (curiously, only augmented by vaccination in those who had pre-vaccine AD-2 antibody) correlate with reduction of viremia in some vaccine studies, e.g., Baraniak, doi: 10.1093/infdis/jiy102. gB-based candidate vaccines have yet to elicit robust responses against this region and a substantial proportion of patients do not make AD-2 antibody. Lantto showed years ago that residues in the first complementarity determining region of both the heavy and the light chain were involved in determining the AD-2 specificity and that key mutations in the germ-line sequence were required for effective interaction with this epitope. Human germ-line IGHV3-30 and IGKV3-11 genes were the only V genes that participated in an AD-2 specific Ab response. He proposed that the inability of the human germ-line gene-encoded Ab repertoire to directly recognize this antigenic determinant resulted in poor immunogenicity in vivo.

Against this background, Jenks mapped the genealogy of B cells encoding potently neutralizing anti-gB AD-2 antibodies and found that a single amino acid heavy chain mutation, A33N, rarely generated by somatic hypermutation machinery, conferred broad CMV neutralization. Structural studies revealed that this mutation mediated key contacts with the gB AD- 2 epitope.

The scientific flow the paper is impeccable and elegant. Appropriate controls are included, and conclusions are supported by the data. The best, most cutting-edge techniques are used. The methodology and, in general, the interpretations are beyond reproach.

I would like to raise threes for the authors to consider and expound upon. First, the implications of the work – the assertion that the solving of the atomic structure of a gB AD peptide bound by an early B cell precursor mAb may represent the structure of the gB AD linear region capable of eliciting potently neutralizing antibodies in the host – maybe be over-stated. This conclusion may be prematurely speculative. Thus, this reviewer would suggest toning down the conclusions and minimizing the speculation that stabilization of the gB AD peptide in this conformation may be the next key step in the design of a next-generation gB candidate vaccine. Maybe this is true; but the authors should be careful to acknowledge potential pitfalls and uncertainties, maybe even going so far as to add a few sentences to indicate what these pitfalls and challenges may engender.

A second, more broad criticism this reviewer has centers around the putative novelty of the finding. The finding of the single amino acid heavy chain mutation, A33N, is very exciting, but in a sense, isn’t it derivative of what Lantto already described? Previous work showed that induction of the VH A33 residue, in this case a A->D mutation, was critical for high-affinity binding. Putting aside the issue of the patentability of the discovery (not relevant to suitability of the manuscript, but still a bit of a curiosity), what makes this new discovery fundamentally distinct from the Lannto paper? This review believes the authors should comment further on this point. In this context, it’s a bit surprising that the control for the A33N mutation was A33G, and that the A33D was not generated and tested (did the authors perform the experiments described in Figure 4 with A33D? That would have been interesting).

The final point that this reviewer would raise has to do with vaccine design. It is straightforward conceptually (although perhaps not at all “easy”) to envisage how these findings might usher in a re-design of therapeutic monoclonal antibodies. In this context, a brief reference to the failed studies of CMV Ig, both polyclonal pooled sera (cytogam, cytotech) and monoclonals, is worth considering. For example, would the Hughes study (DOI: 10.1056/NEJMoa1913569; please consider citing this paper and commenting!) finally “work this time” with this newly reconfigured moab? But, more importantly, HOW, precisely, would the authors envisage this being generated into an ACTIVE vaccine for gB (as opposed to “passive” vaccination with a moab)? After all, we are born with the VDJ recombinatorial machinery we are born with. How could an immunization be designed that would instruct the vaccinated host on how to generate the proper heavy chain IgG sequence? If the authors are really thinking this exciting new data will inform how we design and administer a vaccine, please tell us how the responses of the recipient’s heavy chain sequences can be so modified.

**Part II – Major Issues: Key Experiments Required for Acceptance**

Reviewer #1: 1. The genealogy of neutralizing AD-2S1 antibodies was studied in a single patient, and the crucial A33N mutation was predicted to be improbably. This raises the question of whether neutralizing antibodies generally require unlikely mutations in most or all individuals. A previous study found that non-germ-line encoded residues are critical for effective antibody recognition of the poorly immunogenic gB AD-2 epitope (Lantto et al., Eur J Imunol 2002, PMID 12115649). Are these mutations also predicted to be improbable?

Reviewer #2: 1. in this study the authors have measured antibody binding to multiple different gB targets but seem to have missed the 2 most physiological.

Firstly, they have not assessed binding to virion (unless i have missed it) which is the main target of the antibody. Of course they measure functional activity by neutralisation but the data presented suggests the higher affinity of the antibody for AD2 is the explanation. I suspect it is true but then the antibodies should bind virions with higher affinity. It is possible they don't and there is another functional explanation - it is worth confirming this.

Secondly, they look at AD2 binding in the context of plasma membrane - with actually a very modest effect. Thus AD2 in membranes may be presented differently (including virion membranes) and so is relevant to the point above. Secondly, they have never looked at gB expressed by HCMV (i.e. in infected cells) - it is transfected gB. There is likely a lot of differences in an infected cell that could change gB on the surface compared to transfected gB. I do not suggest it as a replacement for transfection but as an addition.

2. Epithelial cell neutralisation data - I think using the AD169 repaired UL131 virus really is not the virus to use here. A clinical isolate with epithelial cell tropism should be used. Again, we are learning that genes in ULb' could modulate virion content etc and this again could affect the biology of gB. It seems strange given all we have learnt about CMV genetics versions of AD169 virus are still used beyond HFFs.

Reviewer #3: The only experiment I would consider-although this reviewer does not view it as essential-would be to repeat the assays described in Figure 4, but with an A33D mutation in addition to the A33G control.

**Part III – Minor Issues: Editorial and Data Presentation Modifications**

Reviewer #1: 2. The authors talk a lot about B cell lineage-targeted vaccine design without clearly explaining how it works. The concept should be explained to the reader in sufficient detail. It is important for the reader to understand that the mutation frequency within B cells cannot be changed (i.e., an unlikely mutation will remain unlikely), but tweaking the selection of suitable B cell lineages might be possible.

3. HCMV gB has four neutralizing antigenic domains. If it is so difficult to elicit neutralizing antibodies against AD-2S1, wouldn’t it be better to focus on one of the other domains rather than using the B cell lineage-directed approach proposed here?

4. Parts of the discussion are somewhat repetitive. The term “established the feasibility of B cell lineage-based vaccine design” is used very often throughout the manuscript, and it is an overstatement (see general comment above). This statement needs to be toned down.

Reviewer #2: 1. Identity of transfected gB - whilst gB AD2 site 1 is conserved there are still different clades of gB with nearby mutations. have the authors tried other gBs?

2. Whilst the data clearly support A33N as a key event are the other mutations completely uneccesary. Were multiple mutations made? Whilst not required they may make A33N better? Discussion point?

Reviewer #3: No concerns.

PLOS authors have the option to publish the peer review history of their article (what does this mean?). If published, this will include your full peer review and any attached files.

Reviewer #1: No

Reviewer #2: No

Reviewer #3: No

Figure Files:

Data Requirements:

Reproducibility:

References:

---

## [Decision Letter · Decision Letter 1]

9 Jan 2023

Dear Dr. Permar,

We are pleased to inform you that your manuscript 'A single, improbable B cell receptor mutation confers potent neutralization against cytomegalovirus' has been provisionally accepted for publication in PLOS Pathogens.

Best regards,

Robert F. Kalejta

Academic Editor

PLOS Pathogens

Blossom Damania

Section Editor

PLOS Pathogens

Kasturi Haldar

Editor-in-Chief

PLOS Pathogens

orcid.org/0000-0001-5065-158X

Michael Malim

Editor-in-Chief

PLOS Pathogens

orcid.org/0000-0002-7699-2064

Reviewer Comments (if any, and for reference):

Reviewer's Responses to Questions

**Part I - Summary**

Reviewer #1: The authors have responded adequately to my questions and concerns, and I am satisfied with their response.

This paper should be of great interest to the readers of PLoS Pathogens.

Reviewer #2: This study is a resubmission of an article that identified a previously identified mutation using in silico approaches and has gone onto define the impact of this mutation on the activity of AD2 antibodies. Overall, the revisions (both experimental and textual) make the article stronger and more reflective of the state of the art prior and post their own studies.

**Part II – Major Issues: Key Experiments Required for Acceptance**

Reviewer #1: (No Response)

Reviewer #2: I have no major issues

**Part III – Minor Issues: Editorial and Data Presentation Modifications**

Reviewer #1: (No Response)

Reviewer #2: I do disagree with the assertion it isn't possible to generate sufficient wt HCMV for neutralisation experiments. A number of labs have worked on BAC systems and long term culture in epithelial cells to generate stocks that can be used precisely for these types of studies - it is just a lot more time consuming and technically more difficult than using the AD169 repaired virus.

The AD169-UL131 repair virus is not wild type. It is a lab strain that can infect epithelial cells but we do not know if the mechanism is the same as for true wild type virus. However, for the purposes of this study I think it is fine to report these data as long as the AD169-UL131 is appropriately recognised for what it is - a lab strain with engineered epithelial tropism (e.g. in methods).

PLOS authors have the option to publish the peer review history of their article (what does this mean?). If published, this will include your full peer review and any attached files.

Reviewer #1: No

Reviewer #2: No

---

## [Editor Report · Acceptance letter]

17 Jan 2023

Dear Dr. Permar,

We are delighted to inform you that your manuscript, "A single, improbable B cell receptor mutation confers potent neutralization against cytomegalovirus," has been formally accepted for publication in PLOS Pathogens.

Best regards,

Kasturi Haldar

Editor-in-Chief

PLOS Pathogens

orcid.org/0000-0001-5065-158X

Michael Malim

Editor-in-Chief

PLOS Pathogens

orcid.org/0000-0002-7699-2064